# High-Dose Intravenous Ascorbate in Sepsis, a Pro-Oxidant Enhanced Microbicidal Activity and the Effect on Neutrophil Functions

**DOI:** 10.3390/biomedicines11010051

**Published:** 2022-12-25

**Authors:** Kritsanawan Sae-khow, Awirut Charoensappakit, Direkrit Chiewchengchol, Asada Leelahavanichkul

**Affiliations:** 1Center of Excellence on Translational Research in Inflammation and Immunology (CETRII), Department of Microbiology, Chulalongkorn University, Bangkok 10330, Thailand; 2Immunology Unit, Department of Microbiology, Chulalongkorn University, Bangkok 10330, Thailand

**Keywords:** vitamin C, sepsis, neutrophil, neutrophil extracellular traps

## Abstract

Vitamin C (ascorbic acid), a water-soluble essential vitamin, is well-known as an antioxidant and an essential substrate for several neutrophil functions. Because of (i) the importance of neutrophils in microbial control and (ii) the relatively low vitamin C level in neutrophils and in plasma during stress, vitamin C has been studied in sepsis (a life-threatening organ dysfunction from severe infection). Surprisingly, the supraphysiologic blood level of vitamin C (higher than 5 mM) after the high-dose intravenous vitamin C (HDIVC) for 4 days possibly induces the pro-oxidant effect in the extracellular space. As such, HDIVC demonstrates beneficial effects in sepsis which might be due to the impacts on an enhanced microbicidal activity through the improved activity indirectly via enhanced neutrophil functions and directly from the extracellular pro-oxidant effect on the organismal membrane. The concentration-related vitamin C properties are also observed in the neutrophil extracellular traps (NETs) formation as ascorbate inhibits NETs at 1 mM (or less) but facilitates NETs at 5 mM (or higher) concentration. The longer duration of HDIVC administration might be harmful in sepsis because NETs and pro-oxidants are partly responsible for sepsis-induced injuries, despite the possible microbicidal benefit. Despite the negative results in several randomized control trials, the short course HDIVC might be interesting to use in some selected groups, such as against anti-biotic resistant organisms. More studies on the proper use of vitamin C, a low-cost and widely available drug, in sepsis are warranted.

## 1. Introduction

Sepsis is one of the important healthcare problems worldwide with an approximate incidence of 3 per 1000 cases with more than 50% mortality in septic shock [1]. The initial release of massive proinflammatory factors, such as cytokines (IL-1β, IL-6, and TNF-α), reactive oxygen species (ROS), leukocyte-activated chemokines, and nitric oxide, with the altered expression of adhesion molecules in sepsis resulting in multiorgan dysfunctions [2,3]. In contrast, the release of anti-inflammatory mediators, such as IL-10, prostaglandins, soluble proteins, IL-1 receptor antagonist (IL-1ra), immunomodulatory hormones, inhibitors of pathogen recognition, and several immune suppressor cells are counteracting against the overwhelming inflammation that can induce sepsis immune exhaustion (the increased susceptibility against secondary infection after sepsis) [3]. Hence, the balance in sepsis immune responses is complex as the anti-inflammation is necessary for the control of sepsis-hyperinflammation and the boost-up of immune responses might be beneficial during immune exhaustion [4,5,6]. In contrast, the enhanced anti-inflammation during sepsis-induced immune exhaustion and the facilitated immune activation during sepsis hyperinflammation might be harmful to the patients. However, current sepsis treatment is still based on the microbial control (antibiotics and removal of the causes of infection) and maintaining hemodynamic stability (fluid administration and vasopressors), but not the intervention on immune response manipulations. Nevertheless, the research topic of sepsis adjuvant therapy that interferes with the balance of immune responses is ongoing and recent reports using vitamin C, a low-cost widely available drug, in the physiologic concentrations or the supra-physiologic concentration by the high-dose intravenous vitamin C (HDIVC) in sepsis is mentioned [7,8]. Here, we present a review of HDIVC focusing on the possible effect in enhanced microbicidal activities and the influence on neutrophils, the immune cells that need vitamin C for several functions.

## 2. The Kinetics of Vitamin C

Vitamin C (ascorbic acid) is an essential micronutrient in a lactone (C_6_H_8_O_6_) structure with hydroxyl groups (pK values of 4.17 and 11.57) that consists of the reduced form (ascorbate) and the oxidized molecule (dehydroascorbic acid; DHA) [9]. Vitamin C in the reduced form is easily oxidized and destroyed by oxygen, light, alkali, and high temperature which have been generally used as a potent anti-oxidant [9]. Plasma levels of vitamin C reflect dietary intake and typically vary between 20–80 µmol/L; however, the active uptake by leukocytes in the blood circulation and in the tissues results in intracellular concentrations that are 10–100-fold higher than in the plasma [8,10]. Kidneys or livers are the organs where the majority of animal species endogenously produce vitamin C but the ability to produce vitamin C in several mammals (including humans, gorillas, monkeys, bats, and guinea pigs), birds, and fish has been lost due to the inability to produce the L-Gulonolactone oxidase (GLO) enzyme in the final step of vitamin C synthesis [9]. As such, GLO use-up oxygen molecules to catalytically convert L-Gulono-1-lactone into L-Ascorbic acid with hydrogen peroxide (H_2_O_2_) [11,12,13,14]. Although there are several hypotheses regarding the evolutionary advantage of the missing ascorbate biosynthesis in humans, avoidance of unnecessary H_2_O_2_ production during vitamin C synthesis with an effective reabsorption process might partly be the answers [15]. As such, the highly effective uptake of DHA via the glucose transporter-1 (GLUT-1) receptor in red blood cells that only occurs in the species with the lost vitamin C synthesis, the rapid intracellular reduction that rapidly alters DHA into the reduced form (ascorbate) using glutathione (GSH) with the 10-fold more effective reduction than the nicotinamide adenine dinucleotide phosphate (NADPH) pathway might be a compensation for the human loss of vitamin C synthesis [15]. With this effective vitamin C cycling through GLUT-1, the animal species without vitamin C synthesis require vitamin C as minimum as 2–3 mg/kg/day [16], while the species with active vitamin C synthesis through GLO using erythrocyte glucose transporter-4 (Glut-4) require vitamin C up to 200–300 mg/kg/day [17].

Then, dietary vitamin C is essential for humans that is absorbed through sodium-dependent vitamin C transporters 1 (SVCT1) at the apical side of small intestinal enterocytes (SVCT1 is expressed only in the intestines, proximal convoluted renal tubules, and livers) [13]. Due to the easily oxidized nature of vitamin C, ascorbate in the intestinal lumens might be converted into DHA (the oxidized form) that will be absorbed via glucose transporter 2 (GLUT2) and GLUT8 at the apical part of the enterocytes [18]. After the absorption, vitamin C distributes in the water compartment and is taken up in most cells by SVCT2 and several GLUTs, except for red blood cells (RBC) and proximal tubular cells that use GLUT-1 (absorption of the DHA form) and SVCT1 (absorption of the ascorbate form), respectively [19]. In RBC, DHA is rapidly intracellularly altered into ascorbate, referred to as the “ascorbate recycling process” (absorbed ascorbate in the blood is extracellularly oxidized into DHA, DHA in blood is transported into the RBC, and DHA is immediately turned back into ascorbate inside the RBC) is well established [20]. The ascorbate absorption in RBC might be interfered with through some impacts on the glucose transporter (GLUT) such as hyperglycemia in diabetes or acute illness [21]. The plasma vitamin C concentrations lower than 11 µM (the normal range is 50 to 80 µM) is associated with scurvy, while the concentrations in several organs are higher with the millimolar levels and the level in the saliva is very less. In the steady state of vitamin C intake, vitamin C in monocyte, lymphocyte, platelet, and pituitary gland is nearly 3 mM, while close to 1.5–2.0 mM in neutrophil and adrenal gland [19]. Vitamin C concentrations are often low in aging or illnesses (acute and critical conditions), such as myocardial infarction, acute pancreatitis, and sepsis [22,23]. Due to the saturation of the intestinal SVCT1 for the vitamin C transportation, the maximum plasma level after a long-term high-dose oral vitamin C is only 220 µM; however, the intravenous injection can induce the higher plasma pharmacologic concentrations (more than 5 mM) that can reach up to 15 mM [24]. Because one of the most important functions of vitamin C is to support the immune system, all immune cells have active vitamin C transporter molecules embedded in their membranes that actively pump the vitamin into the cells when the higher vitamin C is required [25,26]. Due to the important role of vitamin C in several metabolic processes (especially the DNA and protein synthesis), the benefits of vitamin C in human diseases, including sepsis, cancer, atherosclerosis, diabetes, and neurodegenerative diseases, are mentioned [26,27].

## 3. Pro-Oxidant Vitamin C Versus Warburg Effect of the Immune Cells in Sepsis

Vitamin C plays an important role in protecting cells from several intracellular oxidative stress by maintaining the intracellular redox balance using ascorbate as an electron donor neutralizing several reactive oxygen species (ROS), including superoxide anions, hydroxyl radicals, singlet oxygen, and hypochlorous acid that are naturally generated during several physiologic or pathologic processes (such as metabolic respiration and mitochondrial oxidative phosphorylation) [28,29]. In contrast, HDIVC induces pharmacologic concentrations (high than 5 mM) in the blood leading to an increase in hydrogen peroxide (H_2_O_2_), a potent ROS, that can be used as an adjuvant for cancer treatment. With oral administration, vitamin C levels are tightly controlled by limited enterocyte absorption, saturated tissue transporters, and renal function (reduced reabsorption and increased excretion) [30]. Meanwhile, the intravenous ascorbate induces hundreds of folds higher level than the oral ascorbate depending on the doses and infusion time resulting in the vitamin C pharmacologic concentrations which are the pro-drug for H_2_O_2_ in the extracellular space [31]. Because of the water solubility, pharmacologic ascorbic acid achieves the equivalent concentrations between blood and extracellular fluid and easily sends the electrons to metal protein ions, such as the alteration of Fe^3+^ into Fe^2+^ (oxidation process). Then, the oxidized metal ion (Fe^2+^) donates the electrons to an oxygen molecule forming active oxygen (superoxide), changing into H_2_O_2_ (superoxide combines with a water molecule) that are inhibited in blood by RBC membrane-reducing proteins and several large molecular weight plasma proteins. However, the reducing molecules in the extracellular space are less abundant than in the intravascular compartment allowing positive H_2_O_2_ activities only in the extracellular space (insufficient reducing compounds to neutralize H_2_O_2_) [31]. Due to the activities of ROS, the advantage of pharmacologic ascorbate-induced H_2_O_2_ are frequently mentioned in cancer adjuvant therapy and some of the antimicrobial strategies [32].

The overwhelming H_2_O_2_ with a limited anti-oxidant in the tumor environment facilitates the death of the malignant cells [33]. Interestingly, most of the energy production from cancer cells is from “aerobic glycolysis” (the glycolysis activity despite the adequate oxygen molecules), but not mitochondrial oxidative phosphorylation (OXPHOS; the mitochondria-dependent cell respiration), referred to as “Warburg effect” [34]. Indeed, the adenosine triphosphates (ATPs) from glycolysis activity are predominant in 56–63% of the total ATP production resulting in high glucose consumption and profound lactate production in patients with cancer [34]. The glycolytic predominance in malignant cells is due to the suppression of OXPHOS, but not the defects in mitochondrial function [35], which is different from the “Warburg effect” in the innate immune cells (and, perhaps, also other cells) during sepsis [36]. In sepsis, it is possible that the prominent anaerobic glycolysis is an adaptation against an inadequate ATP from the OXPHOS due to the mitochondrial damages during relative hypoxia that is very common in sepsis-induced hypoperfusion from several factors [37]. In sepsis, there is an imbalance of the oxidant/antioxidant substances in favor of the oxidants, referred to as “oxidative stress” from several processes, such as mitochondrial breakdown and nitric oxide synthesis, with the relatively low anti-oxidant levels [38,39]. Because mitochondria are the major intracellular sources of oxidative substances, partly due to the possible leakage of superoxide from the electron transport chain [40], the reduced OXPHOS in cancer cells and in several cells during sepsis might be an adaptation to reduce oxidative substances. While the cancer cells intentionally avoid the production of oxidants to enhance the cell survival using aerobic glycolysis, the cells during sepsis are forced to enter anaerobic glycolysis due to the mitochondrial damage. Perhaps, the “Warburg effect” in both cancer and sepsis might be an adaptation to reduce oxidative stress through a natural avoidance of mitochondrial injury by switching to glycolysis activity and allowing mitochondrial rest in both conditions. The reduced oxidative stress in cancer is beneficial for tumor cell growth as the oxidants might reduce the malignant cell survival, while a decrease in oxidative stress during sepsis limits further production of the oxidants. Hence, the pro-oxidant effect of HDIVC might differently alter the directions of these natural adaptations in these two conditions. While HDIVC-induced H_2_O_2_ might enhance the elimination of cancer cells, the too prominent peroxide by HDIVC in sepsis might be harmful to the sepsis-induced organ injury and HDIVC should be limited as a short-course administration during sepsis.

## 4. Vitamin C as a Pro-Oxidant against Microbial Agents

Although benefits of the pro-oxidant effects of HDIVC in cancers through an enhanced killing activity against the malignant cells has become clearer, the increased oxidative stress by HDIVC in sepsis is in debate. On one hand, the H_2_O_2_ in extracellular space might enhance microbicidal activities which have been demonstrated in several organisms. For example, the high dose of ascorbate with a very low dose of amphotericin B in mice with sepsis-induced renal injury shows better survival with less fungal burdens in the internal organs than the treatment with amphotericin B alone [41]. This is, perhaps, through the influx of vitamin C-induced H_2_O_2_ via the amphotericin-mediated pores on the fungal cell surface [41]. As such, the pharmacological ascorbate results in an effective fungal control even with the amphotericin B concentrations at 10 times lower than the minimal inhibitory concentration (MIC) which might be used for reducing the side effects amphotericin B in higher doses. A similar synergistic effect is also demonstrated in the use of ceftriaxone (a beta-lactam antibiotic) on multidrug-resistant *Staphylococci* [42] and in anti-tuberculosis [43,44]. Additionally, a clinical study in sepsis exhibited that the HDIVC at 200 mg/kg/day for 4 days (using 50 mg/kg/dose, every 6 h) shows benefit in lower sequential organ failure assessment (SOFA) scores than the placebo in patients [45]; however, the benefit of HDIVC on the microbicidal activity is not mentioned in most clinical trials. On the other hand, there is a debate on the worsening of oxidative stress in sepsis due to the increased pro-oxidants from HDIVC in sepsis. The requirement of vitamin C is increased during infection as up to 3 g daily of vitamin C is needed to restore normal plasma concentrations in patients with sepsis, while only 0.1–0.3 g of vitamin C per day is sufficient in the normal condition [23,46]. With HDIVC (approximately 12 g per day for 4 days), vitamin C levels are elevated 20 to 500 folds from the physiologic level with the benefit of reduced organ failure and 28-day mortality, without the significant side effects, partly due to the short course of the administration [47,48]. However, the concerns about hyperoxaluria and calcium-oxalate stone, especially in patients with a renal impairment which is common in patients with sepsis, and the possible enhanced oxidative stress are raised with HDIVC [49]. In our experiments, there is no oxalate crystallization in the mouse kidneys after a short course of the supra-physiologic vitamin C administration [41].

Perhaps, other impacts of vitamin C, such as endothelial barrier attenuation, improved norepinephrine and vasopressin functions (vitamin C is a cofactor of dopamine β-hydroxylase and peptidyl glycine α-amidating monooxygenase enzymes), and the restored functions of glucocorticoid receptor, might be responsible for the beneficial effects of vitamin C in any concentrations [50,51]. Notably, endogenous stress-induced steroids also increase cellular vitamin C uptake [51]. Although the direct microbicidal activity of the pro-oxidant HDIVC as an adjuvant to bactericidal membrane pore-forming agents is possible, the effect of vitamin C on immune cells, especially neutrophils, might also be important. Several studies revealed that vitamin C deficiency reduces cellular and humoral immune responses [52,53]. In clinical studies, vitamin C treatment of healthy subjects (physiologic concentrations) not only promotes neutrophil functions, but also enhanced natural killer cell activities, and lymphocyte proliferation [54]. Furthermore, vitamin C also stimulated murine immune cells, primarily dendritic cells, to the more distinct interleukin (IL)-12 secretion that possibly activates T and B cell functions [54,55]. However, the studies on these topics with the concentrations of vitamin C in millimolar (the supra-physiologic level) are still limited. Some of the different effects of vitamin C in the physiologic versus high doses of vitamin C is demonstrated in Figure 1.

## 5. Neutrophils in Sepsis

Re-evaluating the influences of sepsis on neutrophil functions and the signal pathways may help to develop new and promising treatment strategies for sepsis. Neutrophils work as double-edged swords during sepsis, including the microbicidal activities versus the enhanced tissue injury (hyper-inflammatory responses) [56]. Multiple mechanisms of sepsis-induced impaired functions of neutrophils, including microbicidal activity, migration, phagocytosis, and neutrophil extracellular traps (NETs), are mentioned as the intracellular organismal killing of neutrophils depends on vitamin C-associated reactive oxygen species (ROS) [57,58,59]. As such, neutrophil extracellular traps (NETs) or “NETosis” is a release of the webs-like structure (NETs) consisting of histone, chromatin, and microbicidal proteins to kill pathogens extracellularly that is triggered by several factors as categorized into the conventional suicidal NETs and the vital NETosis [60,61]. Conventional suicidal NETosis requires phorbol-12-myristate-13-acetate (PMA) and other initiators to release calcium from the endoplasmic reticulum into the cytoplasm that enhances the protein kinase C (PKC) activity and phosphorylation of gp91phox resulting in (i) NADPH oxidase assembly into the functional complexes at phagosome membranes (also called phagocytic oxidase; PHOX) for ROS generation [60], (ii) transfer of myeloperoxidase (MPO) and neutrophil elastase (NE) in the azurophilic granules to the nucleus to degrade histone protein and promote chromatin de-condensation [62], (iii) activation of peptidyl arginine deiminase 4 (PAD4) that leads to histone citrullination [60], and (iv) ROS-induced rupture of the cell membrane [60,62]. For vital NETosis, the neutrophils eject some parts of DNA with 3 different factors to the suicidal NETosis [60], including (i) the preserved membranes integrity using the vesicular DNA trafficking [60], (ii) the preserved intracellular killing capability [63], and (iii) differences in the stimuli and the timing of NETs release [64]. In sepsis, NETs reduce microbial abundance at the early phase of infection [65,66] correlating with the inflammatory cytokine synthesis, while inducing several organ injuries during the killing processes [67]. Additionally, NETs enhanced thrombus formations via (i) the activation of intrinsic coagulation pathway through factor XII (FXII) [68], (ii) the inhibition of several anti-thrombin factors, such as tissue factor pathway inhibitor (TEPI), antithrombin III, and activated protein C (APC), by several NETs components, especially neutrophil elastase (NE) [69], and (iii) the stimulation of the extrinsic coagulation pathway through the tissue factor-bearing NETs [70], resulting in disseminated intravascular coagulation (DIC) in small blood vessels with inadequate tissue perfusion and organ failure [2]. Hence, NETs are beneficial in microbial control in sepsis but might be responsible for organ injuries and DIC leading to other fatal complications of sepsis [68].

## 6. Vitamin C and Neutrophil Functions

Overall, the effect on neutrophil functions in many randomized controlled trials (RCTs) of the supplementation by vitamin C alone, or in combination with other micronutrients or antioxidants, in patients is rarely mentioned, in part, because there is no routine clinical laboratory for neutrophil functions [71,72,73]. The regular routine complete blood count (CBC) indicates only the quantity of neutrophils in blood. Then, data on the impact of vitamin C on neutrophils are mostly from the experimental studies. Similar to RBC, neutrophils also have “ascorbic acid recycling” that can take advantage of both ascorbate and DHA, the reduced and oxidized forms of vitamin C, respectively [19]. The intracellular ascorbate (a negative-charged anion) can act as antioxidant, while the oxidized form (DHA) might restore the ROS and NO production for the microbicidal activity. The interchangeability between ascorbate and DHA is due to the balance of the cell in the oxidative or reduction state with vitamin C as a balance control of the cellular redox homeostasis [74]. Activated neutrophils accumulate vitamin C at 10-fold greater than normal neutrophils via SVCT2 (for ascorbate transport) and GLUTs (GLUT1 and GLUT3 for DHA transport) [14,75]. Notably, GLUTs are more effective vitamin C transporters than SVCT2 [75]. During stress and infection, vitamin C concentration in plasma and neutrophils is rapidly declining, due to the use against the invading microorganisms [76,77]. Indeed, clinical studies demonstrate that vitamin C is necessary for neutrophil chemotaxis, phagocytosis, oxidative burst, NO generation, apoptosis, and NETosis [73,78,79]. Vitamin C can improve neutrophil migration and is also used as a substrate for (i) NADPH oxidase complex to generate superoxide radical reaction for oxidative burst (formation of H_2_O_2_ and chlorinated oxidants) and (ii) nitric oxide (NO) for the killing activity [80]. Moreover, vitamin C is necessary for the proper spontaneous neutrophil apoptosis which is a natural process of neutrophils at 12–24 h after releasing into blood circulation [81,82]. Without proper apoptosis, the vitamin C-deficient neutrophils that lack an appropriate antimicrobial activity will turn into necrotic cells with enhanced systemic inflammation [81].

Regarding NETs formation, there is an increase in NETosis in patients with sepsis, and the ex vivo stimulation of ascorbate at 1 mM or lower (physiologic concentrations) reduced NETosis, while ascorbate that higher than 5 mM (pharmacologic concentrations) facilitate NETosis [79]. As such, ROS and oxidative stress induce NETs, especially through NADPH oxidase (NOX)-dependent mechanism [83], through the stimulation of PAD4, a NETs initiation enzyme affecting nuclear chromatin de-condensation by removal of positively charged ions on histone proteins using deamination or citrullination [84]. In parallel, ROS activates mitogen-activated protein kinase (MAPK) signaling pathways that promotes MPO and NE transportation into the nucleus for the NETs induction process [85]. Naturally, NO from the microbial-activated neutrophils can promote ROS generation to destroy the microorganisms both intracellularly (phagocytosis with microbicidal effects) and extracellularly (NETosis) [78,82]. In NO synthesis, vitamin C stabilizes the co-factor tetrahydrobiopterin (BH4) which is a mediator supporting NO production from nitric oxide synthase (NOS) [86]. Indeed, NO is an important mediator that influences the NADPH oxidase on the phagosomes and on the cell membranes to generate oxygen free radicals (ROS; the substrates of the oxidative burst process) that can initiate NETosis [86]. Furthermore, vitamin C also activates nuclear factor-kappa B (NF-κB) which leads to an increase in *PAD4* expression and facilitates the transportation of MPO and NE from the cytosol into the nucleus causing an enhancement in NETs formation [81]. Hence, the low concentration of vitamin C might act as an anti-oxidant through the negative-charged-anion ascorbate that inhibits NETosis [47], while the high vitamin C concentrations might facilitate NETosis via ROS induction [79]. Neutrophils in mice with vitamin C deficiency also demonstrate the upregulated signaling of autophagy (the reuse of the damaged cell parts by conserved degradation of the cell that unnecessary or dysfunctional components through a lysosome-dependent regulated mechanism) compared with the healthy mice [47,87]. However, impacts of the high vitamin C concentrations on other non-NETs processes of cell death (apoptosis and autophagy) of neutrophils during sepsis are still unknown. In contrast, the ability of high doses vitamin C on the induction of both apoptosis and autophagy in several malignant cells is mentioned [88,89,90]. Hence, it seems that the impacts of vitamin C on several cell deaths depend on the ascorbate concentrations which unfortunately too less data. With a vitamin C deficiency level (less than 11 µM), easier neutrophil necrosis and hyper-inflammation by the signaling from necrotic cells are possible [82,91]. Meanwhile, ascorbate concentrations lower than 1 mM (physiologic ranges) possibly reduced NETs-induced hyperinflammation in sepsis, but vitamin C higher than 5 mM (pharmacologic level) seems to facilitate NETosis [79]. By these postulations, HDIVC might enhance NETosis in sepsis which might be beneficial in the effective microbial control (NETs and peroxide-induced bactericidal activities). Indeed, organismal control is still the main strategy of sepsis treatment and antibiotic-resistant bacteria are currently common [92]. However, HDIVC might enhance NETosis with several harmful effects in sepsis, but the HDIVC benefits possibly outperform the adverse effects with a short duration administration in the recent several clinical trials [45].

Due to the reduced ascorbate level during sepsis and stresses [47,93] and the attenuation of several factors with the restoration of physiologic vitamin C concentrations, including capillary blood flow, microvascular barrier function, vasopressor responsiveness, microvasculature, and endothelial injury with safe and be well tolerated in patients [47], vitamin C administration in sepsis is interesting. Likewise, the physiologic vitamin C concentration in patients with sepsis after abdominal surgery also demonstrates the antiapoptotic effect on peripheral blood neutrophils [73]. In supraphysiologic HDIVC, the safety of the procedure in multiple clinical trials with the promising mortality improvement in the recent studies [13] leads to the possibility of HDIVC in real clinical situations, especially with the infection by antibiotic-resistant organisms. The supraphysiologic ascorbate-induced hydrogen peroxide in the extracellular space and enhanced NETs formation might be beneficial in patients infected with microbial resistance organisms. Unfortunately, the data on NETosis and microbial resistance are not available in most studies using HDIVC in sepsis. Here, a conclusion of the possible impact of vitamin C on the neutrophil effector functions in sepsis is postulated in Figure 2.

## 7. Clinical Trials of Vitamin C Supplementation in Sepsis

The recent examinations of HDIVC alone or in combination with thiamine (vitamin B1) and corticosteroids (hydrocortisone, ascorbic acid, and thiamine; HAT therapy), revealed the possible decreased inflammation and end-organ failure in several diseases. The vitamin C infusion is an attractive and low-cost treatment option not only in patients with sepsis, but also in other inflammation-related diseases, such as acute respiratory distress syndromes (ARDS), vascular injury, and cancer [94,95,96]. However, the outcomes of intravenous vitamin C are inconsistent, especially in patients with sepsis. As such, a few randomized controlled trials (RCT), including the ORANGES trial (vitamin C 1.5 g plus hydrocortisone 50 mg every 6 h with thiamine 200 mg every 12 h for 4 days) [97] and the RCT in China (hydrocortisone 50 mg every 6 h for 7 days with vitamin C 1.5 g plus thiamine 200 mg every 12 h for 4 days) [98] demonstrate the efficacy of vitamin C in the improvement of several sepsis outcomes, such as survival rate, SOFA score, vasopressor requirement, time to resolution of shock and lactate levels. Meanwhile, the VITAMINS trial (vitamin C 1.5 g with hydrocortisone 50 mg every 6 h and thiamine 200 mg every 12 h for 3–5 days) [99], CITRIS-ALI (vitamin C 50 mg/kg every 6 h for 4 days) [96], and VICTAS study (vitamin C 1.5 g with thiamine 100 mg and hydrocortisone 50 mg every 6 h for 4 days) [100], do not find any significant improvements between vitamin C intervention versus placebo control. Moreover, the LOVIT trial (vitamin C 50 mg/kg every 6 h for 4 days) [101] indicates that the patients who received intravenous vitamin C had a higher risk of death or persistent organ dysfunction at 28 days than those who received a placebo. These inconsistencies possibly are partly due to the differences in the characteristics of patients, ethnicity, the baseline of vitamin C level, and vitamin C administration protocols (time to start treatment, doses, and length of the administration). Thus, to determine a possible benefit of HDIVC in sepsis, clinical trials for a comparison between HDIVC versus low-dose intravenous vitamin C or control in a similar baseline of vitamin C plasma level might be necessary. However, the adverse effects of HDIVC might also be a concern. As such, there is an increased prevalence of hypernatremia in HAT therapy in sepsis which might be because of the hydrocortisone adverse effects [98] and a few cases of oxalate nephropathy [102] and acute kidney injury (AKI) [103] after HDIVC administration. Despite the possible harmless of a short course of HDIVC, the benefit of vitamin C in sepsis is still uncertain with several potential adverse effects. With several negative RCTs, the use of HDIVC in sepsis in all patients might be inappropriate. However, more data on the impacts of the intravenous vitamin C treatment against antibiotic-resistant bacteria and in the population with relatively low plasma vitamin C levels are needed before the proper solid conclusion on the use of vitamin C in sepsis.

## 8. Conclusions

Despite several theoretically possible benefits of vitamin C in sepsis to reverse the stress-induced relatively cellular vitamin C deficiency and to enhance microbicidal activity (improved phagolysosome functions of neutrophils), several negative RCTs in patients discourage the general use of vitamin C in patients with sepsis. However, more studies of vitamin C administration (in a short course with proper doses) on the selected groups of patients with sepsis might be necessary before discouraging vitamin C administration in all situations of sepsis.

## Figures and Tables

**Figure 1 biomedicines-11-00051-f001:**
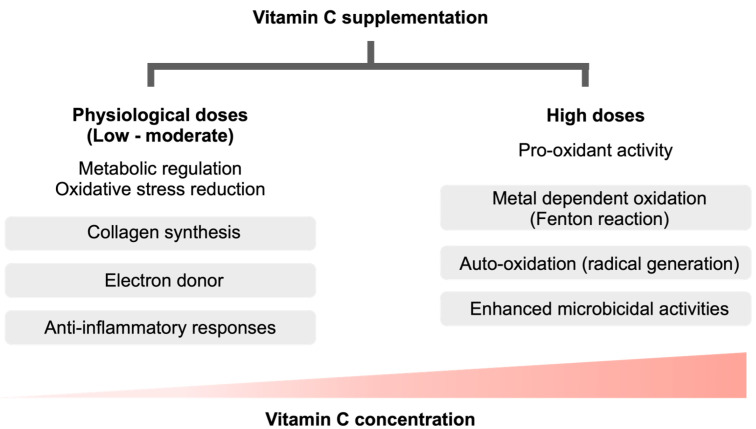
Some of the different benefits and actions of vitamin C between the physiologic concentrations (low to moderate dose administration) versus the supra-physiologic doses. While the physiologic concentrations regulate several metabolic pathways (such as collagen synthesis) and act as an anti-oxidant (an electron donor) with immune modulation effects (anti-inflammatory responses), the supra-physiologic concentrations surprisingly induce pro-oxidant activities; such as metal-dependent oxidation (Fenton reaction) and auto-oxidation (generation of the free radicals) with an enhanced microbicidal activity.

**Figure 2 biomedicines-11-00051-f002:**
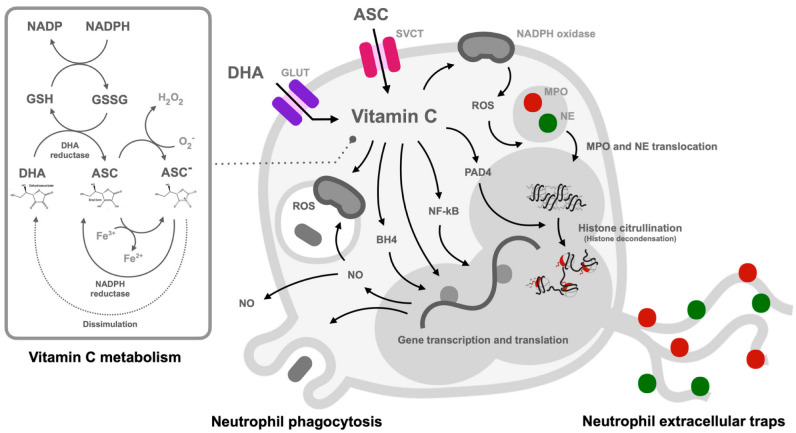
The possible balance in cellular redox homeostasis in neutrophils by ascorbate is demonstrated. The possible kinetics and effectors of vitamin C on the neutrophil functions indicating the entry of ascorbate (ASC; an anti-oxidant) and dehydroascorbic acid (DHA; an oxidized form) through sodium-dependent vitamin C transporter 2 (SVCT2) and glucose transporter (GLUT), respectively. While the physiologic properties of vitamin C can enhance phagocytic activity, oxidative burst, and nitric oxide (NO) synthesis partly through tetrahydrobiopterin (BH4; a cofactor for NO production), vitamin C (especially at the pharmacologic level) might enhance NETosis through the facilitated activities of peptidyl arginine deiminase 4 (PAD4), myeloperoxidase (MPO), and neutrophil elastase (NE) via the reactive oxygen species (ROS) produced through vitamin C-activated nicotinamide adenine dinucleotide phosphate (NADPH) oxidase. Although the oxidized form of vitamin C at the physiologic level facilitates ROS and NO for the intra-neutrophil microbicidal activity, the supraphysiologic vitamin C might facilitate the killing activities with NETosis induction through the enhanced hydrogen peroxide (H_2_O_2_) in the extracellular space. The box on the left side demonstrates an alteration of an oxidized DHA form into an anti-oxidant ASC using glutathione (GSH) and oxidized glutathione (oxidized disulfide; GSSG) with nicotinamide adenine dinucleotide (NAD) and NADPH system. In parallel, ASC can be changed into an anion negative charge (ASC^−^) by the Fenton reaction (Fe^2+^ and Fe^3+^) and oxygen-H_2_O_2_ interaction that cause a pro-oxidant impact. Additionally, a direct change from ASC^−^ into DHA (the dissimulation; in dashed line) of ASC^−^ and DHA is also theoretically possible.

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
