# Peer review of "High-Dose Intravenous Ascorbate in Sepsis, a Pro-Oxidant Enhanced Microbicidal Activity and the Effect on Neutrophil Functions"

_biomedicines, 2022, doi:10.3390/biomedicines11010051_

Round 1

Reviewer 1 Report

Dear colleague Kritsanawan Sae-khow and co-authors, I have read with interest your review on the action of ascorbic acid on neutrophil function. I find it comprehensive in relation to our current understanding of the immunology of inflammation. Further research would be needed to evaluate the potential role of vitamin C for other outcomes in inflammation, vascular injury sepsis and ARDS.

Author Response

Dear, Reviewer

Many thanks for giving us this opportunity. Followed your advices, we agree with all your comments and apologize for the unclear presentation and correct them accordingly. We added section 7, topic of “Clinical trials of vitamin C supplementation in sepsis”, on the clinical trial of HDIVC.

Best regards

Awirut Charoensappakit

Reviewer 2 Report

The authors have completed a comprehensive review of vitamin C supplementation in sepsis. Vitamin C is an attractive, low cost treatment option for patients with sepsis with some physiological basis for presumed benefit. The authors have at length discussed presumed patient benefit of vitamin C, however, they have failed to mention and comment on several trials that failed to show any benefit for patients (CITRIS-ALI, LOVIT, VITAMINS). The protocols used in these studies and their results need to be discussed. Also, the authors should outline more clearly possible physiology behind failures of these trials to show clinical benefit. Accordingly, the conclusion needs to be changed to reflect the outcomes of clinical trials.   

Author Response

Dear, Reviewer

Many thanks for giving us this opportunity. Followed your advices, we agree with all your comments and apologize for the unclear presentation and correct them accordingly. We added section 7, topic of “Clinical trials of vitamin C supplementation in sepsis”, on the clinical trial of HDIVC.

In this section, we provided the results of recent randomized controlled trials of intravenous vitamin C supplementation in patients with sepsis and the possible explanations for discussing their distinct observations. 

Best regards

Awirut Charoensappakit

Round 2

Reviewer 2 Report

I would like to thank the authors for preparing a revised manuscript. However, I still cannot agree with the conclusion, which I find misleading because of at least inconclusive (3 multicentre RCTs), if not negative (1 multicentre RCT) results of clinical trials I think the evidence from clinical trials cannot be used to encourage the use of vitamin C. The authors have eloquently discussed the possible role of low dose and short term therapy with vitamin C, but that should be tested first. 

Author Response

I would like to thank the authors for preparing a revised manuscript. However, I still cannot agree with the conclusion, which I find misleading because of at least inconclusive (3 multicentre RCTs), if not negative (1 multicentre RCT) results of clinical trials I think the evidence from clinical trials cannot be used to encourage the use of vitamin C. The authors have eloquently discussed the possible role of low dose and short term therapy with vitamin C, but that should be tested first.

ANS: We thank the reviewer for the comment and agree on the discouraging of vitamin C use in sepsis. However, we would lie to open the conclusion a little bit for some groups that still lac of the data (antibiotic resistance and low vitamin C in plasma). Hence, we toned down all conclusion in abstract and at the end of manuscript as following;

Abstract: “Despite the negative results in several randomized control trials, the short course HDIVC might be interesting to use in some selected groups; such as against anti-biotic resistant organisms. More studies on the proper use of vitamin C, a low-cost and wildly available drug, in sepsis are warranted.”

Discussion: “Despite the possible harmless of a short course of HDIVC, the benefit of vitamin C in sepsis is still uncertain with several potential adverse effects and several negative RCTs, the use of HDIVC in sepsis in all patients might be inappropriate. However, more data on the impacts of the intravenous vitamin C treatment against antibi-otic-resistant bacteria and in the population with relatively low plasma vitamin C levels are needed before the proper solid conclusion on the use of vitamin C in sepsis.  ”

Conclusion: “Despite several theoretically possible benefits of vitamin C in sepsis to reverse the stress-induced relatively cellular vitamin C deficiency and to enhance microbicidal ac-tivity (improved phagolysosome functions of neutrophils), several negative RCTs in patients discourage the general use of vitamin C in patients with sepsis. However, more studies of vitamin C administration (in a short course with proper doses) on the selected groups of patients with sepsis might be necessary before discouraging vitamin C administration in all situations of sepsis.”
